# Exploring the feasibility and acceptance of an optimised physiotherapy approach for lateral elbow tendinopathy: a qualitative investigation within the OPTimisE trial

Marcus Bateman ![ORCID],[1,2] Benjamin Saunders ![ORCID],[2] Karin Cooper,[3] Chris Littlewood,[4] Jonathan C Hill ![ORCID] [2]

[1]University Hospitals of Derby and Burton NHS Foundation Trust, Derby, UK
[2]Faculty of Medicine & Health Sciences, Keele University, Keele, UK
[3]Patient Representative, Derby, UK
[4]Faculty of Health, Social Care & Medicine, Edge Hill University, Ormskirk, UK

**Correspondence to**
Marcus Bateman;
marcus.bateman@nhs.net

## ABSTRACT

**Objectives** To explore the acceptability of an optimised physiotherapy (OPTimisE) intervention for people with lateral elbow tendinopathy (LET) and feasibility of comparing it to usual care in a randomised controlled trial.

**Design** Semistructured interviews, analysed using thematic analysis and mapped onto the COM-B model of behaviour change.

**Setting** Conducted as part of the OPTimisE Pilot & Feasibility randomised controlled trial within physiotherapy departments in the United Kingdom National Health Service.

**Participants** 17 patients with LET (purposively sampled to provide representativeness based on age, sex, ethnicity, deprivation index and treatment allocation) and all 8 physiotherapists involved as treating clinicians or site principal investigators.

**Results** Four themes were identified. First, participants reported the OPTimisE intervention as acceptable. Second, differences between the OPTimisE intervention and usual care were identified, including the use of an orthosis, holistic advice/education including modifiable risk factors, forearm stretches, general upper body strengthening and a more prescriptive exercise-dosing regimen. Third, participants provided feedback related to the trial resources, which were viewed positively, but identified language translation as a need. Fourth, feedback related to trial processes identified the need for changes to outcome collection and reduction of administrative burden. From the perspective of adopting the OPTimisE intervention, we found evidence that participants were able to change their behaviour. Considering the findings through the lens of the COM-B model, the intervention is likely to be deliverable in practice and the trial can be delivered at scale with some additional support for physiotherapists.

**Conclusions** Overall, the OPTimisE intervention was found to be different to usual care and acceptable to patients and physiotherapists. The study highlighted the need to refine trial processes and resources prior to a full-scale trial, to reduce administrative burden, increase support for physiotherapists, improve return rate of outcome questionnaires and provide language translation.

## STRENGTHS AND LIMITATIONS OF THIS STUDY

⇒ Patient participants from a range of social and ethnic backgrounds contributed to this study.
⇒ These are the opinions of people accessing healthcare for lateral elbow tendinopathy (LET) and so may not reflect the perceptions of those who do not access healthcare for LET, including some underserved groups.
⇒ We were unable to interview some patients who failed to attend their allocated treatment sessions, so may not have captured a full range of perceptions.

**Trial registration number** ISRCTN database 19 July 2021. https://www.isrctn.com/ISRCTN64444585.

## INTRODUCTION

Lateral elbow tendinopathy (LET), also known as tennis elbow, is a common condition affecting the elbow that causes pain and impacts daily function.[1–3] Physiotherapy is recommended if symptoms fail to settle naturally within 6 weeks but there is wide variation in the types of treatments provided by physiotherapists.[4–6] The OPTimisE intervention was designed to be a simple physiotherapy intervention that could be easily implemented. Research evidence was combined with the opinions of expert clinicians, physiotherapy service managers and patients who had experienced LET, to agree what an optimised physiotherapy treatment package should contain.[7] The OPTimisE intervention comprises three elements: condition-specific and general health advice, supported by printed and online resources; a progressive exercise regime working within limits of pain deemed acceptable by individual patients; and the provision of a counterforce orthosis.

This qualitative study was embedded within a two-arm multicentred pilot and feasibility randomised controlled trial (RCT) investigating whether the OPTimisE intervention could be tested against usual care in a real-world healthcare setting.[8] The OPTimisE Pilot & Feasibility Trial opened in September 2021, recruiting 50 patient participants across three UK National Health Service physiotherapy clinics within a 12-month period, as per the published protocol.[8] Patients were randomised to receive the OPTimisE intervention or usual physiotherapy treatment. Patients in the OPTimisE intervention group received a printed manual, login details for a password-protected patient information webpage and an Epi-Hit elbow orthosis. They were given advice on a range of condition-specific and broader health-related topics, and were taught a regime of exercises. Further information regarding the advice topics, exercises and fitting the orthosis was described in the manual and in video format on the webpage. All patients were asked to complete outcome questionnaires at 6-week, 12-week and 6-month follow-up and were given the option of choosing whether this was done online or via return post. All patients were provided with a Squegg device to objectively measure their grip strength and report the values as part of the questionnaire. The device links via Bluetooth to a smartphone application that includes a strength measurement tool and games to promote gripping exercises. Patients in the OPTimisE intervention group received the Squegg device after randomisation and those in the usual care group were sent it by post at 6 months to coincide with their final questionnaire. The number of follow-up appointments was left at the discretion of the treating physiotherapists, who were advised to leave at least 4 weeks between appointments.

In addition to quantitative measures of feasibility, trial participants were interviewed to explore their perceptions and experiences related to the trial design and intervention protocol. We aimed to explore the acceptability of delivering/receiving the OPTimisE intervention and perceptions on the processes employed in the pilot trial to inform feasibility alongside the quantitative results. This study has been reported in line with the COnsolidated criteria for REporting Qualitative research checklist.[9]

Acceptability is defined by Sekhon et al as 'a multifaceted construct that reflects the extent to which people delivering or receiving a healthcare intervention consider it to be appropriate, based on anticipated or experienced cognitive and emotional responses to the intervention'.[10] They propose an Acceptability of Healthcare Interventions Framework describing seven components that can be used to assess the acceptability of an intervention: affective attitude, burden, ethicality, intervention coherence, opportunity costs, perceived effectiveness and self-efficacy. In order to implement a new intervention into real-world practice, there needs to be evidence that clinicians are willing to adapt their behaviour in favour of the new intervention. Patients need to engage with the new intervention, especially when it is designed to empower patients to self-manage their condition, as in the case of the OPTimisE intervention. The COM-B model can be used to assess this behaviour change.[11] The model defines three key components:

1. Capability—the individual's psychological and physical capacity to engage in the activity.
2. Opportunity—factors that lie outside the individual that make the behaviour possible or prompt it.
3. Motivation—brain processes that energise and direct behaviour.

Our key objective, therefore, was to explore participants' perceptions and experiences through the lens of the COM-B model as a means of evidencing changes in behaviour and assess acceptability using Sekhon et al's framework, to provide supportive evidence that the OPTimisE intervention can be adopted into real-world clinical practice and inform a future main trial, as well as highlighting potential changes to the intervention or trial processes.

## METHOD

### Qualitative sampling and recruitment

Patients consenting for the OPTimisE Pilot & Feasibility Trial were asked whether they gave permission to be contacted for an individual interview, following their course of physiotherapy treatment. All completed baseline demographic questionnaires and a core set of baseline measures.[12] Ethnicity options were based on those used in a recent rotator cuff tendinopathy trial, supplemented with a free-text 'other' option for individuals to self-describe.[13] Those who gave permission were purposively sampled to include people with varied ages, sex, ethnicity, deprivation index and treatment allocation within the trial, as far as was possible within the sample recruited to the trial. Patients were sent a letter of invitation by post, accompanied by a participant information sheet (PIS) and followed up by email or telephone 2 weeks later, to ask if they wished to be interviewed. All physiotherapists involved as site principal investigators (PIs) or treating clinicians delivering the OPTimisE intervention were emailed a letter of invitation and PIS, asking them to reply if they wished to volunteer. All participants were given the option of face-to-face, telephone or video-conference calls at a mutually convenient time. All patient participants opted for telephone interviews and all physiotherapist participants video-conference calls. All interviews were audio recorded and all participants provided recorded verbal consent after being read a consent form. All participants were sent a £20 gift voucher to thank them for their time.

### Data collection

All interviews were conducted by MB, a white British middle-aged male consultant physiotherapist and PhD candidate who has qualitative research training, between February 2022 and January 2023. The interviewer was not

known to the patient participants but they were aware that he was the chief investigator for the OPTimisE Pilot & Feasibility Trial. The physiotherapist participants at one trial site have worked with MB and those at other sites knew him from site initiation visits and trial communication/meetings. All participants were encouraged to speak freely about their opinions, whether positive or negative.

Interviews were semistructured, using a topic guide developed by MB and BS (see online supplemental file 1) and reviewed by the patient coinvestigator (KC). The topic guide was iteratively revised based on early analysis. 60 minutes were allocated for each interview but the mean duration for patients was 28 minutes (range 18–42) and physiotherapists 28 min (23–35). Interviews were not repeated. Following the interviews, the recordings were uploaded via an encrypted web portal to an independent transcription service (https://www.universitytranscriptions.co.uk/) to be transcribed verbatim and returned via encrypted download. All transcriptions were checked for accuracy by MB and any uncertainties were resolved by relistening to the original audio recording. Transcripts were not returned to participants as there were no unresolved transcription issues. Patient participant interviews continued until data saturation was reached, whereas physiotherapist participant interviews were limited by the small population, so all participants were interviewed. Saturation was assessed in terms of 'informational redundancy', where new interview data no longer provided fresh insights.[14]

## Data analysis

Anonymised interview transcripts were analysed using inductive thematic analysis.[15] MB coded all of the transcripts using NVivo V.12 software. Codes were explored both within and across interview transcripts, then indexed into areas of relevance, based on patterns within the data, to form provisional codebooks for each participant group. MB, BS, JH and KC then met in person to review the patient participant data and finalise the codebook. MB, BS and JH reviewed the physiotherapist participant data and finalised the codebook, before both codebooks were compared. Codes were grouped according to similar topics (or subthemes) and, from these, themes were developed that overlapped both codebooks. These themes and subthemes were then examined through the lens of the three core components of the COM-B model to provide evidence of behaviour change, deliverability of a main trial and identification of processes that can be improved.[11] Similarly, the codes were mapped to the Acceptability of Healthcare Interventions Framework to provide evidence of intervention acceptability.[10]

## Patient and public involvement

Patient volunteers were involved with the design of the OPTimisE intervention, selection of orthosis, generation of trial website frequently asked questions and review of trial resources. KC is a member of the OPTimisE Patient and Public Involvement Group, contributing to the trial design, analysis of the qualitative data and is a coauthor on this paper.

## RESULTS

From a total of 50 patients recruited to the OPTimisE Pilot & Feasibility Trial, 45 gave permission to be contacted to discuss taking part in a qualitative interview. Following purposive sampling, 24 of these patients were invited to be interviewed and 17 participated. One other patient initially agreed to be interviewed but later changed their mind due to busy work and personal schedules. The other six did not respond to email and telephone follow-up. The median age of patient participants was 47 (range 37–62) with an even split related to sex and treatment group allocation within the trial. Individuals from a range of ethnic and social backgrounds were included, representative of the demographic of the general population. 13 identified as white British, 1 white other, 1 Pakistani, 1 Sri Lankan and 1 Kosovar. Median deprivation score was 6 (range 1–10), where 1 is the highest level of deprivation and 10 is the lowest level of deprivation (The Index of Deprivation measures seven domains: income deprivation; employment deprivation; education, skills and training deprivation; health deprivation and disability; crime; barriers to housing and services; living environment deprivation.), measured in deciles.[16] Median symptom duration was 6 months (range 2–36) and median baseline Patient Reported Tennis Elbow Evaluation score was 47 (range 18.5–93) at the time of recruitment to the OPTimisE trial. Demographic data from patient participants are provided in table 1. In addition, all eight of the site PIs and physiotherapists who delivered the OPTimisE intervention to

| Table 1 | Patient participant demographics | |
|---|---|---|
| **Identifier** | **Age** | **Sex** |
| BHX003 | 47 | Male |
| BHX004 | 47 | Male |
| DER001 | 52 | Male |
| DER002 | 39 | Female |
| DER003 | 54 | Male |
| DER004 | 55 | Female |
| DER006 | 39 | Female |
| DER008 | 54 | Male |
| DER011 | 40 | Female |
| SHE001 | 42 | Female |
| SHE004 | 52 | Female |
| SHE005 | 48 | Female |
| SHE011 | 37 | Male |
| SHE013 | 62 | Male |
| SHE014 | 43 | Female |
| SHE016 | 54 | Male |
| SHE018 | 47 | Male |

**Table 2** Physiotherapist participant demographics

| Identifier | Sex | Grade* |
|---|---|---|
| PT1 | Male | Band 7 |
| PT2 | Male | Band 7 |
| PT3 | Male | Band 6 |
| PT4 | Female | Band 7 |
| PT5 | Female | Band 6 |
| PT6 | Female | Band 6 |
| PT7 | Female | Band 8a |
| PT8 | Male | Band 7 |

*A newly qualified physiotherapist starts at band 5. Band 6 represents a senior role, while band 7 represents a specialist role. Band 8a represents a managerial or highly specialist clinical role.

patients during the trial agreed to be interviewed. They had a range of 7–35 years' experience as a qualified physiotherapist. Additional physiotherapist participant demographics are shown in table 2.

Four themes were identified from the data: experiences of the OPTimisE intervention; differences between the OPTimisE intervention and usual care; feedback related to trial processes and feedback related to the trial resources. The map of themes and related subthemes is displayed in table 3, with detailed coding trees supplied as online supplemental files 23).

### Theme 1: experiences of the OPTimisE intervention

The OPTimisE intervention was received positively by both patients and physiotherapist participants. From the perspective of delivery, physiotherapist participants reported that it was practical to provide patients with the three treatment components within a standard 30-min

session and that the suggested follow-up times of approximately 4 weeks could be accommodated. Some found that follow-up sessions could be performed by telephone, without the need of a face-to-face consultation, given that patients could refer to the visual aids in the patient handbook or trial website. Indeed, due to its comprehensiveness and clarity, some felt that the intervention could be delivered in a single session, with patients advised to self-manage using the resources provided

> It was fairly straightforward to deliver. That was the nice thing about it. I only had a positive experience… The one thing that it really did make me reflect on is just how the information was packaged and how it was brought together and the breadth of the information. That was the really lovely thing about doing it. You did it and just felt why aren't all physiotherapy interventions a bit like this? It's really clear. PT2

> It's almost quite a nice self-management programme… because there was so much information at the start with it, it almost made follow ups a little bit redundant. PT3

In relation to the advice and education component of the OPTimisE intervention, physiotherapist participants fed back that the holistic health content they were asked to teach was familiar, as it was common practice to provide this for certain conditions, such as chronic low back pain, despite not usually providing it for people with LET. Patient participants reported that while not all the topics were relevant to all people, for some the advice resonated, causing them to address certain lifestyle factors. Examples were reducing alcohol and tobacco use, losing weight, increasing general exercise levels and getting more sleep.

**Table 3** Map of themes and subthemes

| Themes | Subthemes |
|---|---|
| Experiences of the OPTimisE intervention | Patient participants' perceptions on the advice and education component |
| | Patient participants' perceptions on the exercise component |
| | Patient participants' perceptions on the orthotic component |
| | Physiotherapist participants' experience of delivering the OPTimisE intervention |
| | Time feasibility for physiotherapist participants |
| Differences between the OPTimisE intervention and usual care | Perceptions on the addition of stretches, exercise selection and dosing |
| | Perceptions on advice/education topics |
| | Perceptions on the use of orthoses |
| Feedback related to trial processes | Patient participants' experience of the outcome questionnaires |
| | Perceptions on patient treatment randomisation |
| | Use of the Squegg device |
| | Physiotherapist participants' perceptions on the site training and trial scalability |
| | Suggestions for improvements |
| Feedback related to the trial resources | Patient participants' feedback on the trial website, participant information sheet and patient manual |
| | Physiotherapist participants' feedback on the intervention manual, patient manual, electronic investigator site file and trial website |

I have cut down quite a lot on smoking, so I am pretty chuffed with myself for doing that. And I don't drink like I used to do because me and my husband did like a drink, but we have both cut down loads, which is good because it's a healthier option I suppose, instead of filling your body full of toxins. SHE005

While the orthosis provided to patients did not help everyone, many reported that it offered short-term pain relief. There were no concerns raised regarding the choice of product. Indeed, some commented that it was superior to others available and the optional wrist support provided additional benefit for some individuals.

Because I started using the elbow strap and when I was lifting things it was helping with the pain—there was no pain. SHE001

They were high quality orthotic devices compared to the sort of things I've seen in the past and participants seem to like them. PT2

Patient participants reported that the exercise component of the intervention could be fitted into their daily routines. There was positive feedback from both patient and physiotherapist participants regarding the simplicity of exercises and exercise progression.

Yes, because they don't take a huge amount of time it's been really easy to kind of fit them into a routine. It's been something that I can do, you know, quite easily and it tends to be when I get back from work, that I tend to do it… If I'm at work, they are exercises I can do quite easily at my desk, if I need to, as well… it's a fairly, I'd say, narrow range of exercise - they build up on each other really well I thought. SHE016

Doing this exercise once a day is quite achievable isn't it, to the patient?… So as a concept I could very much sell it because I believe I could subscribe to that if I was a patient. It was easy to do. PT6

### Theme 2: differences between the OPTimisE intervention and usual care

Physiotherapist participants perceived that usual care for LET lacked consistency and structure, with variations in exercises prescribed and exercise dosing among colleagues within their teams. The inclusion of stretches as part of the OPTimisE intervention was highlighted as something that none of the treating physiotherapists would ordinarily use in their practice. Usual care was thought to centre on progressive loading of the forearm extensor muscles and advice/education based on a mechanical model of pathology understanding. The OPTimisE intervention, in contrast, was perceived to have a biopsychosocial model, incorporating holistic health advice/education with a structured rehabilitation programme, which was regarded positively.

I think it just flags up again what we should be doing as a whole, in regards to all of our MSK [musculoskeletal] patients. Which is the biopsychosocial-type

model of care and not forgetting about the extra bits-and-pieces that go alongside tendon healing, like lifestyle changes and all the rest of it. Like I say, you can in a busy clinic and when you've got not enough time to reflect, it's easy to brush over the other bits-and-pieces, rather than, great, I've got a quick lateral epicondyle pain here. I can just give them a quick loading programme and send them on their way. I think it slowed your processing down and think actually look at the bigger picture here, make sure you're addressing the other symptoms or issues that might be affecting this patient. PT1

The exercise dosing in the OPTimisE intervention was perceived to be clearly prescribed, whereas in usual care dosing practice was described as more varied. The promotion of patient self-efficacy by teaching ways to progress and regress exercise difficulty based on symptom response, extending to high load and global upper limb strengthening, was another difference that was identified by physiotherapist participants. Likewise, the inclusion of advice to increase general cardiovascular exercise, combined with addressing other metabolic lifestyle factors.

For me, the novel thing about it was all of the way it was presented, the structure and having all of that support information all in one place in an easy-to-follow way for patient participants. I thought it brought down a lot of these barriers sometimes you get with communicating with patients and feeling like you've got to cram so much into one session. The fact that this was laid out almost as a programme to follow was the nice thing about delivering it… It was a really high level of support and information around that exercise regime that you were delivering. PT2

The provision of an elbow brace orthosis was not typical of usual care, so this was entirely new to some physiotherapist participants, while others reported that they might suggest that patients purchase their own. Many of the physiotherapist participants said they would now change their practice as a result of participating in the pilot trial.

I've had an interesting conversation with two fireman friends of mine who both use elbow clasps now, because they both asked me my opinion and—I think a year ago, I told them I'd have pooh-poohed it, but now I'll give it a go—and actually both of them are climbing now again with using a clasp. So, I guess it's broadened my horizons slightly to think maybe there's something in this, and if it works for the patient—happy days! PT6

### Theme 3: feedback related to the trial resources

Patient participants reported that the PIS was comprehensive and that they felt sufficiently informed about the pilot trial. One patient participant, who had a mild learning difficulty, suggested that the key information be highlighted and separated from the more detailed

aspects, for example, data protection policy, to make it easier to read.

> If you have like really important info to get over… all the blurb about data protection and if you want your data to be used… just really separating that from the main text you want to get across. SHE011

Feedback related to the patient manual and trial website was consistently positive. The website was used by physiotherapist participants as an additional training resource to initially familiarise themselves with the exercise videos and advice website hyperlinks. Some patient participants did not feel the need to access the website as they found the patient manual sufficiently comprehensive. Those that did reported having found it useful and some had followed the advice from the linked websites. The majority used the patient manual as their main resource, commenting that the descriptions of the exercises, dosing and progression/regression were easy to understand.

> We were given a booklet that tells you all about the exercises that we are going to be doing and it explains them all. And it has a proper good description that if I think that I've forgot, I can then look in the booklet and see the exercise that I have been asked to do and I can just, you know, refresh my brain, which is very handy. SHE001

Having the patient manual, containing details of all the OPTimisE intervention components, in one neatly packaged booklet was perceived by physiotherapist participants to add value, save time and allow follow-up consultations to be conducted by telephone if necessary.

> Yes, I mean the handbook was really good. The website, as well—that was really useful initially to be able to make sure I was doing my exercises correctly… both of them combined have been really useful. It helps that the exercises are fairly simple really and because it is a fairly limited range of exercises, you are not constantly shooting on to a new thing. So, I would say the resources have been really good. SHE016

The only critical feedback of the physiotherapists' intervention handbook was a recommendation to add more detail to the basic science section. A new electronic investigator site file (eISF) system, containing all of the administrative resources, was piloted by the trial sponsor with site PIs finding no technical issues but requesting clearer indexing and a single location for regularly accessed files.

### Theme 4: feedback related to trial processes
#### Patient participant perspectives
The patient participants exhibited a positive attitude towards randomisation and did not express strong preferences for their treatment group allocation. They were enthusiastic about the possibility of receiving a novel treatment that could potentially be more effective than standard care but were equally happy to be randomised to either treatment out of a philanthropic inclination to help other people by contributing to the research. Additionally, they appreciated the opportunity to interact with clinicians who were perceived as experts due to their involvement in research.

The Squegg device, used to measure grip strength, functioned as intended, apart from one person who was unable to get the device to work and some needed assistance from more technology-aware family members to set it up. In addition to measuring their grip strength for the outcome questionnaires, some patients reported that they used the games built into the application as a way of improving strength and others stated that family also used the device.

> I've noticed with the Squegg and the games I play, and then it calculates at the end how many grips you've done and my grips have been like between 300 and 350 at a time, so I think with these little games it helps, and it also gives you a bit of entertainment while you're doing a bit of physiotherapy. DER001

There were mixed opinions regarding the outcome questionnaires and a feeling from some patient participants that there were too many questions, which was burdensome for those with busy work and family commitments. It was proffered that highlighting the monetary incentive for returning the questionnaire might encourage more timely completion. The first follow-up questionnaire was sent 6 weeks after randomisation but some patients had only just, or not yet started their treatment due to waiting lists. It was therefore suggested that the 6-week questionnaire could be removed in a future main trial, although patient participants would be amenable to completing a 12-month follow-up questionnaire.

> I definitely think that would be a good idea because sometimes in six months you can be—for example, for me, I've had no pain so you'd think, Okay, I'm fine now. I'll drop the exercise. I'll just go back to my normal life. But then, I think, with these kinds of things they have a tendency sometimes to come back. So, it would be good to see if it, in terms of your research project, did it come back? DER002

We also proposed the addition of a monthly text message, that was positively received. Patient participants felt that it would act as a reminder that they were still part of the trial, be easy to respond to and provide the trial team with additional insight into how their symptoms might fluctuate over time.

> I think for convenience it's probably easier to yes, just fill in a single answer text… I think the questionnaire was useful because it does focus your mind on, you know, thinking 'Is it feeling better than last time?' because it's a wider spread of questions, so for me, probably, combining the two would work. SHE016

Patient participants were given the option of completing questionnaires online or on paper. Those who opted for paper stated that they could have completed them online

if that were the default method. The majority using the online service had no problems. The only issues raised were a display issue on a smartphone and feedback to improve the wording of the communication email from the third-party provider to make it clearer that it was related to the trial.

### Physiotherapist participant perspectives

From the perspective of the physiotherapist participants, there were no major concerns regarding the trial processes. The site training sessions were well received and deemed to be sufficiently comprehensive. Physiotherapist participants agreed that if the trial were to be delivered at scale, across multiple geographical areas, then the training sessions could be conducted via video conference, provided that all of the site hard resources had been posted in advance. Most physiotherapist participants experienced a gap of several weeks between the intervention training and treating their first trial patient. It was suggested that a 5-min refresher video or treatment process summary sheet could be produced to help remind physiotherapists of what to do, or hosting an online discussion forum where physiotherapists could seek advice from the trial team.

An observation from all three trial sites was that recruitment rates declined over the latter half of the recruitment period. It was speculated that this was due to an increasing number of physiotherapists being employed as first contact practitioners (FCPs are healthcare practitioners, typically physiotherapists, who work autonomously in UK Primary Care to assess, diagnose and treat musculoskeletal conditions, fulfilling a role historically performed by General Practice physicians.) in primary care nationally, with patients with LET managed more in community settings, rather than in hospital outpatient physiotherapy services. It was therefore suggested that a future main trial target these clinical settings as the intervention, being well resourced and straightforward to deliver, could be easily delivered in primary care.

I mean this would lend itself really nicely to an FCP clinic because you've got all the information out there for the patient to use and access for self-management so that they'd need to know about it as well. PT5

I think this sort of programme is ideal for that sort of FCP land[scape] It's simple. They can signpost them straight to it, teach them the exercises quite quickly and manage these patients in primary care probably. PT8

The only other feedback related to reducing some of the administrative burden placed on the site PIs by transferring the responsibility of minimum data telephone calls and eISF document monitoring to the trial team. It was also highlighted that for a future trial, provision should be considered for language translation to widen accessibility for underserved patient groups.

## DISCUSSION

In this study, we investigated the acceptability of the OPTimisE intervention and pilot trial processes, to determine whether it is feasible to conduct a full-scale clinical trial. We found the OPTimisE intervention to be deliverable in a publicly-funded healthcare setting and patients engaged with it. The quality of the resources provided to patients was viewed positively, and deemed to add value compared with usual care. The OPTimisE intervention was found to differ from usual care in four important aspects: the provision of an orthosis, holistic advice/education regarding biopsychosocial influences on pain, addition of forearm stretches and general upper body strengthening, and a more prescriptive exercise dosing regimen that included progression or regression based on symptom response. It was suggested that the OPTimisE intervention could be delivered at a single clinic visit with patients encouraged to self-manage using the resources as a guide. There were no concerns regarding the processes of patient recruitment, randomisation or treatment delivery but changes to outcome measure collection will need to be incorporated into a main trial design. These include reducing the length of the outcome questionnaire, removal of the 6 week and addition of a 12-month follow-up questionnaire, incentivisation of all questionnaires and addition of a monthly text message question. In a full-scale trial, the intervention training could be delivered remotely but required the addition of a walk-through checklist

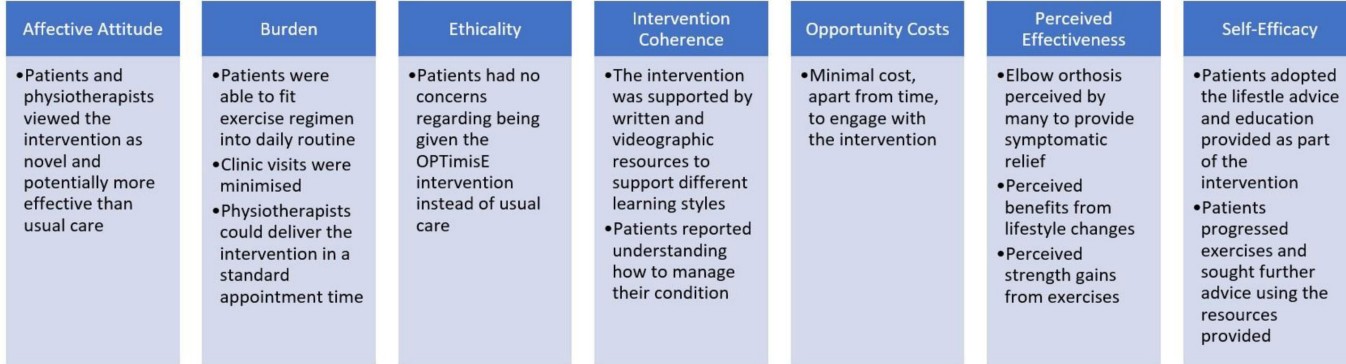

**Figure 1** Mapping to the acceptability of healthcare interventions framework.

or refresher video to help physiotherapists prepare for their first trial patient consultations. The administrative burden on physiotherapists could also be reduced by reassignment of some duties to the trial team. Language translation will also need to be incorporated, to reach underserved communities. It was also proposed that a future trial might be sited in a primary care setting.

The inclusion of qualitative research in feasibility studies is now recognised as an important method of gaining additional insight into how an intervention or trial processes may be improved and consequently increase the impact of a main trial. O'Cathain *et al* describe a range of questions that can be used in a feasibility study for an RCT, the majority of which were applicable to this study, particularly around the subjects of intervention delivery, trial processes, selection of outcomes and completion of outcome measures.[17] We have used these to identify important differences between the OPTimisE intervention and usual care that will allow for a meaningful comparison in a real-world trial, and highlight processes within the trial design that require refinement.

### Implementation

From the perspective of implementing the OPTimisE intervention, we found evidence that patient and physiotherapist participants were able to change their behaviour. If we consider the findings through the lens of the three core components in the COM-B model (capability, motivation and opportunity), we can demonstrate that the intervention is likely to be deliverable in practice and that the trial can be delivered at scale with some additional support for physiotherapists.[11]

### Capability

Patient participants were able to engage with the OPTimisE intervention but physiotherapist participants found that there was an initial period of learning to adapt their practice before they were able to deliver the intervention efficiently. This could be mitigated by providing them with a training refresher video or walk-through checklist in a future full-scale trial.

### Motivation

Patient participants were motivated to take part in the trial as the intervention was perceived as something new and potentially more effective than usual care, with access to specialist physiotherapists involved in research. There was also a sense of taking part for the benefit of the greater good, in order to help other people with LET. Physiotherapist participants appeared motivated by learning new skills and provided evidence that they had adopted some of the treatment components of the OPTimisE intervention into their practice beyond their involvement in the trial.

### Opportunity

Patient participants were involved in the design of the OPTimisE intervention in an attempt to maximise the opportunity for engagement with the intervention. The

exercise programme was simplified to take a maximum of 15 min/day, follow-up clinic visits were kept to a minimum of 4 weeks in-between visits if required, and resources were presented in hardcopy and online formats with written, pictorial and videographic content to suit a variety of learning styles. Feedback was therefore positive from the patient participants interviewed but opportunity could be widened by targeting underserved communities and translating resources into other languages. The intervention was also designed during the COVID-19 pandemic to be deliverable remotely and we found evidence of follow-up consultations being delivered by telephone, for convenience of patients and physiotherapists.

### Acceptability

Study findings about the acceptability of the OPTimisE intervention and feasibility of comparing it to usual care in an RCT are presented in figure 1, mapped to the constructs within the Acceptability of Healthcare Interventions Framework. All seven domains have been satisfied, suggesting that the OPTimisE intervention was acceptable.

### Strength and limitations

Strengths of this study are that it included individuals from a range of backgrounds and used established models/frameworks to assess behaviour change and acceptability. We acknowledge though that these interviews are the opinions of people accessing healthcare for LET and so may not reflect the perceptions of those who do not access healthcare for LET, including some underserved groups. We were also unable to interview some patients who failed to attend their allocated treatment sessions, as they did not respond to interview invitations, so may not have captured a full range of perceptions.

## CONCLUSION

Overall, the OPTimisE intervention was found to be different to usual care and acceptable to both patient and physiotherapist participants. The study highlighted the need to refine trial processes and resources prior to a full-scale trial, to reduce administrative burden, provide additional support for physiotherapists, improve the return rate of outcome questionnaires and provide language translation.

**Acknowledgements** The authors wish to thank all the participants who gave up their time to be interviewed. MB would like to thank the British Elbow & Shoulder Society for the award of a Research Pump-Priming Grant used to support preliminary study design and application for Fellowship funding.

**Contributors** MB conducted the interviews and the transcript coding, with mentorship from BS. MB produced the initial codebook before MB, BS, CL, KC and JCH collectively finalised the codebook and interpreted the themes. All authors contributed to this final written report. MB was responsible for the overall content as the guarantor.

**Funding** Marcus Bateman is funded by a National Institute for Health Research (NIHR) and Chartered Society of Physiotherapy Charitable Trust Doctoral Fellowship (reference NIHR300704).

**Competing interests**  None declared.

**Patient and public involvement**  Patients and/or the public were involved in the design, or conduct, or reporting, or dissemination plans of this research. Refer to the Methods section for further details.

**Patient consent for publication**  Not applicable.

**Ethics approval**  Approvals were received from the Yorkshire & The Humber—Sheffield Research Ethics Committee (reference 21-YH-0121) and the UK Health Research Authority (reference 297637) on 22 June 2021. Participants gave informed consent to participate in the study before taking part.

**Provenance and peer review**  Not commissioned; externally peer reviewed.

**Data availability statement**  Data are available upon reasonable request. A copy of the codebooks is available on written request.

**ORCID iDs**
Marcus Bateman http://orcid.org/0000-0002-3203-506X
Benjamin Saunders http://orcid.org/0000-0002-0856-1596
Jonathan C Hill http://orcid.org/0000-0001-6246-1409

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
