## [Reviewer comments · BMJ Open]

ARTICLE DETAILS

TITLE (PROVISIONAL)	Exploring the Feasibility and Acceptance of an Optimised Physiotherapy Approach for Lateral Elbow Tendinopathy: A Qualitative Investigation within the OPTimisE Trial.
AUTHORS	Bateman, Marcus; Saunders, Benjamin; Cooper, Karin; Littlewood, Chris; Hill, Jonathan

VERSION 1 – REVIEW

REVIEWER	Erel , Suat Pamukkale Universitesi
REVIEW RETURNED	06-Jul-2023

GENERAL COMMENTS	First of all, thank you for your contribution to the literature. I would like to say that I like the design of your work and your article. Such remote methods for treatment and evaluation bring many advantages. But I think that these methods will be difficult to use for some disadvantaged groups (with low cognitive level, etc.). Have you encountered such a group during the study? In your article, information about the minimum conditions necessary for the use of this method by individuals can be given.
--

REVIEWER	Paraskevopoulos , Eleftherios University of West Attica
REVIEW RETURNED	01-Aug-2023

GENERAL COMMENTS	Congratulation on your research and thank you for the opportunity to revise this manuscript. I think the major problems found in this paper are the lack of a clear description of your methods and statistical analysis. I think that a paragraph that clearly explains what the optimised intervention does/includes should be included. Also, in qualitative studies the number of participants in the study depends on data saturation. You have not discussed this at all. Please write clearly what your aims are and re-read your manuscript prior to submission.
--

REVIEWER	Leirós-Rodríguez, Raquel Universidad de Leon - Campus de Ponferrada
REVIEW RETURNED	08-Nov-2023

GENERAL COMMENTS	First of all, I would like to congratulate you on your research. It deals with an interesting and highly relevant object of study, while at the same time applying a methodology that brings additional value to traditional quantitative methodology. However, the paper presents methodological limitations and formal errors that should be corrected before its possible publication in this
---

	Journal. ABSTRACT: Abbreviations are not encouraged in this section, please remove them. INTRODUCTION: I consider the Introduction on the pathology of study, to be vague and too succinct. Please, independently of the justification of the components of the intervention applied by you, explain which is the most frequent and validated physiotherapeutic intervention to date (doi: 10.3233/BMR-210053). However, the authors reverse the introduction to develop the methodology. It makes NO sense and is not appropriate. Please provide the Background of the subject of the question in a proper way. MATERIAL AND METHODS AND RESULTS Correct DISCUSSION This section again suffers from an approach that is too concrete and, above all, does not use bibliographical references relevant to the field of study. Please rewrite, augment and give more content and meaning to this section.
--	--

REVIEWER	Murphy, Myles Edith Cowan University, Nutrition and Health Innovation Research Institute
REVIEW RETURNED	23-Nov-2023

GENERAL COMMENTS	Dear Dr Bateman and colleagues, Congratulations on a well conducted qualitative study, for the clear compliance to an accepted qualitative study guideline (COREQ) and in genuinely attempting to get a diverse sample of patient participants. I have provided some suggestions below for your consideration: Abstract  1. Suggest defining OPTImisE. Introduction  2. Page 5: Suggest amending the word 'views' to perceptions – I think perceptions is the more commonly used in qualitative research description. Methods  3. Can the authors reference the source for the ethnicity options? Was this self-reported or from a standardised list. Just as ethnicity can be a controversial and good to support your selection of responses. 4. Page 6: MB is described a male – which refers to sex and not gender. I would suggest gender identity is more relevant to the aims of qualitative research than sex and man/woman/non-binary are preferentially used. 5. Page 6: I would also include other details about MB such as age, ethnicity, whether or not this was performed as a part of a research degree. 6. The title for patient and public involvement seems to differ to the rest of the manuscript? Suggest amending if not purposeful. 7. The methods section would benefit from additional citations to support the methodology (e.g., referencing COREQ and other
--

	methodological papers that support your approach). I do not think more context references are needed, these are well done. Results 8. Based on the results, participant sex seems to have been reported, not gender as the title suggests. Suggest amending in the results and methods were it states gender was collected. 9. For readers outside the UK, I would suggest including what “Band” means in the table of physiotherapist practitioners. 10. Suggest amending terminology throughout to patient participant or physiotherapy participant to differentiate your study participants from patients and physiotherapists more generally.
--	--

VERSION 1 – AUTHOR RESPONSE

Reviewer: 1

Dr. Suat Erel , Pamukkale Universitesi

Comments to the Author:

First of all, thank you for your contribution to the literature. I would like to say that I like the design of your work and your article. Such remote methods for treatment and evaluation bring many advantages. But I think that these methods will be difficult to use for some disadvantaged groups (with low cognitive level, etc.). Have you encountered such a group during the study? In your article, information about the minimum conditions necessary for the use of this method by individuals can be given.

Thank you for the comments. We have included a full range of patient diversity in this sample. As stated in the demographic information, this includes Index of Deprivation from 1 to 10. A footnote has been added to explain the seven domains that the Index of Deprivation measures. This included ‘education, skills and training deprivation’. A reference to The English Indices of Deprivation 2019 Research Report has also been added.

We also provided patient information in different formats (e.g. hardcopy, for those without internet access) although there was no funding for language translation. This is acknowledged as a limitation in the discussion section.

Reviewer: 2

Dr. Eleftherios Paraskevopoulos , University of West Attica

Comments to the Author:

Congratulation on your research and thank you for the opportunity to revise this manuscript. I think the major problems found in this paper are the lack of a clear description of your methods and statistical analysis. I think that a paragraph that clearly explains what the optimised intervention does/includes should be included. Also, in qualitative studies the number of participants in the study depends on data saturation. You have not discussed this at all. Please write clearly what your aims are and re-read your manuscript prior to submission.

Thank you for the comments. Please can I draw your attention to the introduction, where the optimised intervention is described: “The OPTimisE intervention comprises three elements: condition-specific and general health advice, supported by printed and online resources; a progressive exercise regime working within limits of pain deemed acceptable by individual patients; and the provision of a counter-force orthosis.”

We agree with the recommendation to include information regarding data saturation. This has now been added:

“Patient participant interviews continued until data saturation was reached, whereas physiotherapist participant interviews were limited by the small population, so all participants were interviewed. Saturation was assessed in terms of ‘informational redundancy’, where new interview data no longer provided fresh insights.¹¹”

Further clarification has been provided regarding the aims and objectives:

“Our key objective, therefore, was to explore participants’ views and experiences through the lens of the COM-B model as a means of evidencing changes in behaviour and assess acceptability using Sekhon et al.’s framework, to provide supportive evidence that the OPTimisE intervention can be adopted into real-world clinical practice and inform a future main trial, as well as highlighting potential changes to the intervention or trial processes.”

Reviewer: 3

Dr. Raquel Leirós-Rodríguez, Universidad de Leon - Campus de Ponferrada

Comments to the Author:

Dear Authors,

First of all, I would like to congratulate you on your research. It deals with an interesting and highly relevant object of study, while at the same time applying a methodology that brings additional value to traditional quantitative methodology.

However, the paper presents methodological limitations and formal errors that should be corrected before its possible publication in this Journal.

ABSTRACT:

Abbreviations are not encouraged in this section, please remove them.

We have reduced the number of abbreviations to the two most commonly used, but feel that expanding all abbreviations will be detrimental as it will result in removal of important detail, due to word limit constraints. We will, of course, do so if the journal guidelines do not permit abbreviations.

INTRODUCTION:

I consider the Introduction on the pathology of study, to be vague and too succinct. Please, independently of the justification of the components of the intervention applied by you, explain which is the most frequent and validated physiotherapeutic intervention to date (doi: 10.3233/BMR-210053).

The focus of this paper is about the results of our clinical trial’s qualitative findings. Elsewhere, we have published a detailed description of the most frequently-used and evidence-based interventions that contributed to the development of our OPTimise trial intervention (also published in BMJ Open - <http://dx.doi.org/10.1136/bmjopen-2021-053841>). Whilst we note the reviewer’s own recent paper is of interest, we don’t see that it is particularly appropriate or required, within the remit of this current paper to re-visit that detailed summary, or comment specifically on the reviewer’s paper.

However, the authors reverse the introduction to develop the methodology. It makes NO sense and is not appropriate. Please provide the Background of the subject of the question in a proper way.

We take your point and have therefore moved the section describing the OPTimisE Pilot & Feasibility Trial to the background section, to improve the structure and flow of the paper.

MATERIAL AND METHODS AND RESULTS

Correct

DISCUSSION

This section again suffers from an approach that is too concrete and, above all, does not use bibliographical references relevant to the field of study.

Please rewrite, augment and give more content and meaning to this section.

We acknowledge that the references used do not directly relate to lateral elbow tendinopathy research, but they do relate directly to the methods used in the development and testing of a complex clinical intervention, such as the OPTimisE intervention, and how the feasibility of implementing such an intervention can be assessed qualitatively. We reference Michie et al's COM-B model of behaviour change which is highly relevant to this intervention; the Sekhon Acceptability Framework of Healthcare Interventions, which is directly related to the objectives of this qualitative study; O'Cathain et al's guidance on how qualitative studies should be conducted within feasibility trials.

Reviewer: 4

Dr. Myles Murphy, Edith Cowan University

Comments to the Author:

Dear Dr Bateman and colleagues,

Congratulations on a well conducted qualitative study, for the clear compliance to an accepted qualitative study guideline (COREQ) and in genuinely attempting to get a diverse sample of patient participants. I have provided some suggestions below for your consideration:

Abstract

1. Suggest defining OPTimisE.

This has been done.

Introduction

2. Page 5: Suggest amending the word 'views' to perceptions – I think perceptions is the more commonly used in qualitative research description.

This has been done

Methods

3. Can the authors reference the source for the ethnicity options? Was this self-reported or from a standardised list. Just as ethnicity can be a controversial and good to support your selection of responses.

This has been added: "All completed baseline demographic questionnaires and a core set of baseline measures.¹² Ethnicity options were based upon those used in a recent rotator cuff tendinopathy trial, supplemented with a free-text 'other' option for individuals to self-describe.¹³"

4. Page 6: MB is described a male – which refers to sex and not gender. I would suggest gender identity is more relevant to the aims of qualitative research than sex and man/woman/non-binary are preferentially used.

I have deliberately described myself as male and would prefer not to engage with this gender debate.

5. Page 6: I would also include other details about MB such as age, ethnicity, whether or not this was performed as a part of a research degree.

Now described as: "a white British middle-aged male consultant physiotherapist and PhD candidate who has qualitative research training"

6. The title for patient and public involvement seems to differ to the rest of the manuscript? Suggest amending if not purposeful.

This is a requirement of the journal.

7. The methods section would benefit from additional citations to support the methodology (e.g., referencing COREQ and other methodological papers that support your approach). I do not think more context references are needed, these are well done.

We have now added a statement regarding COREQ and also a statement about data saturation, both with references.

“This study has been reported in line with the COnsolidated criteria for REporting Qualitative research checklist (Ref).”

“Patient participant interviews continued until data saturation was reached, whereas physiotherapist participant interviews were limited by the small population, so all participants were interviewed. Saturation was assessed in terms of ‘informational redundancy’, where new interview data no longer provided fresh insights (Ref).”

Results

8. Based on the results, participant sex seems to have been reported, not gender as the title suggests. Suggest amending in the results and methods were it states gender was collected.

The headers of the demographics tables have been changed from gender to sex, to clarify this.

9. For readers outside the UK, I would suggest including what “Band” means in the table of physiotherapist practitioners.

A footnote has been added, to explain the meaning of the bands.

“ A newly qualified physiotherapist starts at Band 5. Band 6 represents a senior role, whilst band 7 represents a specialist role. Band 8a represents a managerial or highly specialist clinical role.”*

10. Suggest amending terminology throughout to patient participant or physiotherapy participant to differentiate your study participants from patients and physiotherapists more generally.

This has been done.